# Differentially Private Change-Point Detection

**Rachel Cummings**
Georgia Institute of Technology
rachelc@gatech.edu

**Sara Krehbiel**
University of Richmond
krehbiel@richmond.edu

**Yajun Mei**
Georgia Institute of Technology
ymei@gatech.edu

**Rui Tuo**
Texas A&M University
ruituo@tamu.edu

**Wanrong Zhang**[*]
Georgia Institute of Technology
wanrongz@gatech.edu

## Abstract

The *change-point detection problem* seeks to identify distributional changes at an unknown change-point $k^*$ in a stream of data. This problem appears in many important practical settings involving personal data, including biosurveillance, fault detection, finance, signal detection, and security systems. The field of *differential privacy* offers data analysis tools that provide powerful worst-case privacy guarantees. We study the statistical problem of change-point detection through the lens of differential privacy. We give private algorithms for both online and offline change-point detection, analyze these algorithms theoretically, and provide empirical validation of our results.

## 1 Introduction

The *change-point detection problem* seeks to identify distributional changes at an unknown change-point $k^*$ in a stream of data. The estimated change-point should be consistent with the hypothesis that the data are initially drawn from pre-change distribution $P_0$ but from post-change distribution $P_1$ starting at the change-point. This problem appears in many important practical settings, including biosurveillance, fault detection, finance, signal detection, and security systems. For example, the CDC may wish to detect a disease outbreak based on real-time data about hospital visits, or smart home IoT devices may want to detect changes in activity within the home. In both of these applications, the data contain sensitive personal information.

The field of *differential privacy* offers data analysis tools that provide powerful worst-case privacy guarantees. Informally, an algorithm that is $\epsilon$-differentially private ensures that any particular output of the algorithm is at most $e^\epsilon$ more likely when a single data entry is changed. In the past decade, the theoretical computer science community has developed a wide variety of differentially private algorithms for many statistical tasks. The private algorithms most relevant to this work are based on the simple output perturbation principle that to produce an $\epsilon$-differentially private estimate of some statistic on the database, we should add to the exact statistic noise proportional to $\Delta/\epsilon$, where $\Delta$ indicates the *sensitivity* of the statistic, or how much it can be influenced by a single data entry.

We study the statistical problem of change-point problem through the lens of differential privacy. We give private algorithms for both online and offline change-point detection, analyze these algorithms theoretically, and then provide empirical validation of these results.

---

[*]Primary author. Authors are listed in alphabetical order.

## 1.1 Related work

The change-point detection problem originally arose from industrial quality control, and has since been applied in a wide variety of other contexts including climatology [LR02], econometrics [BP03], and DNA analysis [ZS12]. The problem is studied both in the *offline setting*, in which the algorithm has access to the full dataset $X = \{x_1, \ldots, x_n\}$ up front, and in the *online setting*, in which data points arrive one at a time $X = \{x_1, \ldots\}$. Change-point detection is a canonical problem in statistics that has been studied for nearly a century; selected results include [She31, Pag54, Shi63, Rob66, Lor71, Pol85, Pol87, Mou86, Lai95, Lai01, Kul01, Mei06, Mei08, Mei10, Cha17].

Our approach is inspired by the commonly used Cumulative Sum (CUSUM) procedure [Pag54]. It follows the generalized log-likelihood ratio principle, calculating

$$\ell(k) = \sum_{i=k}^{n} \log \frac{P_1(x_i)}{P_0(x_i)}$$

for each $k \in [n]$ and declaring that a change occurs if and only if $\ell(\hat{k}) \geq T$ for MLE $\hat{k} = \operatorname{argmax}_k \ell(k)$ and appropriate threshold $T > 0$. The existing change-point literature works primarily in the asymptotic setting when $k_n^*/n \to r$ for some $r \in (0, 1)$ as $n \to \infty$ (see, e.g., [Hin70, Car88]). In contrast, we consider finite databases and provide the first accuracy guarantees for the MLE from a finite sample ($n < \infty$).

In offering the first algorithms for *private* change-point detection, we primarily use two powerful tools from the differential privacy literature. REPORTMAX [DR14] calculates noisy approximations of a stream of queries on the database and reports which query produced the largest noisy value. We instantiate this with partial log-likelihood queries to produce a private approximation of the the change-point MLE in the offline setting. ABOVETHRESH [DNR$^+$09] calculates noisy approximations of a stream of queries on the database iteratively and aborts as soon as a noisy approximation exceeds a specified threshold. We extend our offline results to the harder online setting, in which a bound on $k^*$ is not known a priori, by using ABOVETHRESH to identify a window of fixed size $n$ in which a change is likely to have occurred so that we can call our offline algorithm at that point to estimate the true change-point.

## 1.2 Our results

We use existing tools from differential privacy to solve the change-point detection problem in both offline and online settings, neither of which have been studied in the private setting before.

**Private offline change-point detection.** We develop an offline private change-point detection algorithm OFFLINEPCPD (Algorithm 1) that is accurate under one of two assumptions about the distributions from which data are drawn. As is standard in the privacy literature, we give accuracy guarantees that bound the additive error of our estimate of the true change-point with high probability. Our accuracy theorem statements (Theorems 2 and 4) also provide guarantees for the non-private estimator for comparison. Since traditional statistics typically focuses on the the asymptotic consistency and unbiasedness of the estimator, ours are the first finite-sample accuracy guarantees for the standard (non-private) MLE. As expected, MLE accuracy decreases with the sensitivity of the measured quantity but increases as the pre- and post-change distribution grow apart. Interestingly, it is constant with respect to the size of the database. In providing MLE bounds alongside accuracy guarantees for our private algorithms, we are able to quantify the cost of privacy as roughly $D_{KL}(P_0||P_1)/\epsilon$.

We are able to prove $\epsilon$-differential privacy under the first distributional assumption, which is that the measured quantity has bounded sensitivity $\Delta(\ell)$, by instantiating the general-purpose REPORTMAX algorithm from the privacy literature with our log-likelihood queries (Theorem 1). Importantly and in contrast to our accuracy results, the distributional assumption need only apply to the hypothesized distributions from which data are drawn; privacy holds for arbitrary input databases. We offer a limited privacy guarantee for our second distributional assumption, ensuring that if an individual data point is drawn from one of the two hypothesized distributions, redrawing that data from either of the distributions will not be detected, regardless of the composition of the rest of the database (Theorem 3).

**Private online change-point detection.** In ONLINEPCPD (Algorithm 2), we extend our offline results to the online setting by using the ABOVETHRESH framework to first identify a window in which the change is likely to have happened and then call the offline algorithm to identify a more precise approximation of when it occurred. Standard $\epsilon$-differential privacy under our first distributional assumption follows from composition of the underlying privacy mechanisms (Theorem 5).[2] Accuracy of our online mechanism relies on appropriate selection of the threshold that identifies a window in which a change-point has likely occurred, at which point the error guarantees are inherited from the offline algorithm (Theorem 6).

**Empirical validation.** Finally, we run several Monte Carlo experiments to validate our theoretical results for both the online and offline settings. We consider data drawn from Bernoulli and Gaussian distributions, which satisfy our first and second distributional assumptions, respectively. Our offline experiments are summarized in Figure 1, which shows that change-point detection is easier when $P_0$ and $P_1$ are further apart and harder when the privacy requirement is stronger ($\epsilon$ is smaller). Additionally, these experiments enhance our theoretical results, finding that OFFLINEPCPD performs well even when we relax the assumptions required for our theoretical accuracy bounds by running our algorithm on imperfect hypotheses $P_0$ and $P_1$ that are closer together than the true distributions from which data are drawn. Figure 2 shows that ONLINEPCPD also performs well, consistent with our theoretical guarantees.

## 2  Preliminaries

Our work considers the statistical problem of change-point detection through the lens of differential privacy. Section 2.1 defines the change-point detection problem, and Section 2.2 describes the differentially private tools that will be brought to bear.

### 2.1  Change-point background

Let $X = \{x_1, \ldots, x_n\}$ be $n$ real-valued data points. The *change-point detection problem* is parametrized by two distributions, $P_0$ and $P_1$. The data points in $X$ are hypothesized to initially be sampled i.i.d. from $P_0$, but at some unknown change time $k^* \in [n]$, an event may occur (e.g., epidemic disease outbreak) and change the underlying distribution to $P_1$. The goal of a data analyst is to announce that a change has occurred as quickly as possible after $k^*$. Since the $x_i$ may be sensitive information—such as individuals' medical information or behaviors inside their home—the analyst will wish to announce the change-point time in a privacy-preserving manner.

In the standard non-private offline change-point literature, the analyst wants to test the null hypothesis $H_0 : k^* = \infty$, where $x_1, \ldots, x_n \sim_{\text{iid}} P_0$, against the composite alternate hypothesis $H_1 : k^* \in [n]$, where $x_1, \ldots, x_{k^*-1} \sim_{\text{iid}} P_0$ and $x_{k^*}, \ldots, x_n \sim_{\text{iid}} P_1$. The log-likelihood ratio of $k^* = \infty$ against $k^* = k$ is given by

$$\ell(k, X) = \sum_{i=k}^{n} \log \frac{P_1(x_i)}{P_0(x_i)}. \tag{1}$$

The maximum likelihood estimator (MLE) of the change time $k^*$ is given by

$$\hat{k}(X) = \operatorname{argmax}_{k \in [n]} \ell(k, X). \tag{2}$$

When $X$ is clear from context, we will simply write $\ell(k)$ and $\hat{k}$.

An important quantity in our accuracy analysis will be the Kullback-Leibler distance between probability distributions $P_0$ and $P_1$, defined as $D_{KL}(P_1||P_0) = \int_{-\infty}^{\infty} P_1(x) \log \frac{P_1(x)}{P_0(x)} dx = \mathbb{E}_{x \sim P_1}[\log \frac{P_1(x)}{P_0(x)}]$. We always use $\log$ to refer to the natural logarithm, and when necessary, we interpret $\log \frac{0}{0} = 0$.

We will measure the additive error of our estimations of the true change point as follows.

**Definition 1** (($\alpha, \beta$)-accuracy)**.** *A change-point detection algorithm that produces a change-point estimator $\tilde{k}(X)$ where a distribution change occurred at time $k^*$ is $(\alpha, \beta)$-accurate if $\Pr[|\tilde{k} - k^*| < \alpha] \geq 1 - \beta$, where the probability is taken over randomness of the algorithm and sampling of $X$.*

## 2.2 Differential privacy background

Differential privacy bounds the maximum amount that a single data entry can affect analysis performed on the database. Two databases $X, X'$ are *neighboring* if they differ in at most one entry.

**Definition 2** (Differential Privacy [DMNS06]). *An algorithm $\mathcal{M} : \mathbb{R}^n \to \mathcal{R}$ is $(\epsilon, \delta)$-differentially private if for every pair of neighboring databases $X, X' \in \mathbb{R}^n$, and for every subset of possible outputs $\mathcal{S} \subseteq \mathcal{R}$,*

$$\Pr[\mathcal{M}(X) \in \mathcal{S}] \leq \exp(\epsilon) \Pr[\mathcal{M}(X') \in \mathcal{S}] + \delta.$$

*If $\delta = 0$, we say that $\mathcal{M}$ is $\epsilon$-differentially private.*

One common technique for achieving differential privacy is by adding Laplace noise. The *Laplace distribution* with scale $b$ is the distribution with probability density function: $\text{Lap}(x|b) = \frac{1}{2b} \exp\left(-\frac{|x|}{b}\right)$. We will write $\text{Lap}(b)$ to denote the Laplace distribution with scale $b$, or (with a slight abuse of notation) to denote a random variable sampled from $\text{Lap}(b)$.

The *sensitivity* of a function or query $f$ is defined as $\Delta(f) = \max_{\text{neighbors } X, X'} |f(X) - f(X')|$. The Laplace Mechanism of [DMNS06] takes in a function $f$, database $X$, and privacy parameter $\epsilon$, and outputs $f(X) + \text{Lap}(\Delta(f)/\epsilon)$. Our algorithms rely on two existing differentially private algorithms, REPORTMAX [DR14] and ABOVETHRESH [DNR+09], which are overviewed in Appendix A. Appendix B covers the concentration inequalities used in the proofs of our bounds.

## 3 Offline private change-point detection

In this section, we investigate the differentially private change point detection problem in the setting that $n$ data points $X = \{x_1, \ldots, x_n\}$ are known to the algorithm in advance. Given two hypothesized distributions $P_0$ and $P_1$, our algorithm OFFLINEPCPD privately approximates the MLE $\hat{k}$ of the change time $k^*$. We provide accuracy bounds for both the MLE and the output of our algorithm under two different assumptions about the distributions from which the data are drawn, summarized in Table 1.

Table 1: Summary of non-private and private offline accuracy guarantees under $H_1$. The expressions $\Delta(\ell)$, $A_\delta$, $C$, and $C_M$ are defined in (4), (5), (8), (9), resp.

| Assumption | MLE | OFFLINEPCPD |
|---|---|---|
| $A := \Delta(\ell) < \infty$ | $\frac{2A^2}{C^2} \log \frac{32}{3\beta}$ | $\max\left\{\frac{8A^2}{C^2} \log \frac{64}{3\beta}, \frac{4A}{C\epsilon} \log \frac{16}{\beta}\right\}$ |
| $A := A_\delta < \infty$ | $\frac{67}{C_M^2} \log \frac{64}{3\beta}$ | $\max\left\{\frac{262}{C_M^2} \log \frac{128}{3\beta}, \frac{2A \log(16/\beta)}{C_M \epsilon}\right\}$ |

The first assumption essentially requires that $P_1(x)/P_0(x)$ cannot be arbitrarily large or arbitrarily small for any $x$. We note that this assumption is not satisfied by several important families of distributions, including Gaussians. The second assumption, motivated by the $\delta > 0$ relaxation of differential privacy, instead requires that the $x$ for which this log ratio exceeds some bound $A_\delta$ have probability mass at most $\delta$.

Although the accuracy of OFFLINEPCPD only holds under the change-point model's alternate hypothesis $H_1$, it is $\epsilon$-differentially private for any *hypothesized* distributions $P_0, P_1$ with finite $\Delta(\ell)$ and privacy parameters $\epsilon > 0, \delta = 0$ *regardless of the distributions from which $X$ is drawn*. We offer a similar but somewhat weaker privacy guarantee when $\Delta(\ell)$ is infinite but $A_\delta$ is finite, which roughly states that a data point sampled from either $P_0$ or $P_1$ can be replaced with a fresh sample from either $P_0$ or $P_1$ without detection.

### 3.1 Offline algorithm

Our proposed offline algorithm OFFLINEPCPD applies the report noisy max algorithm [DR14] to the change-point problem by adding noise to partial log-likelihood ratios $\ell(k)$ used to estimate the change

point MLE $\hat{k}$. The algorithm chooses Laplace noise parameter $A/\epsilon$ depending on input hypothesized distributions $P_0, P_1$ and privacy parameters $\epsilon, \delta$ and then outputs

$$\tilde{k} = \underset{1 \leq k \leq n}{\operatorname{argmax}}\{\ell(k) + Z_k\}. \tag{3}$$

Our algorithm can be easily modified to additionally output an approximation of $\ell(\tilde{k})$ and incur $2\epsilon$ privacy cost by composition.

---

**Algorithm 1** Offline private change-point detector : OFFLINEPCPD$(X, P_0, P_1, \epsilon, \delta, n)$

---

    **Input:** database $X$, distributions $P_0, P_1$, privacy parameters $\epsilon, \delta$, database size $n$
    **if** $\delta = 0$ **then**
        Set $A = \max_x \ \log \frac{P_1(x)}{P_0(x)} - \min_{x'} \ \log \frac{P_1(x')}{P_0(x')}$               # set $A = \Delta\ell$ as in (4)
    **else**
        Set $A = \min\{t \ : \ \max_{i=0,1} \Pr_{x \sim P_i}[2|\log \frac{P_1(x)}{P_0(x)}| > t] < \delta/2\}$      # set $A = A_\delta$ as in (5)
    **end if**
    **for** $k = 1, \dots, n$ **do**
        Compute $\ell(k) = \sum_{i=k}^{n} \log \frac{P_1(x_i)}{P_0(x_i)}$
        Sample $Z_k \sim \text{Lap}(\frac{A}{\epsilon})$
    **end for**
    Output $\tilde{k} = \underset{1 \leq k \leq n}{\operatorname{argmax}}\{\ell(k) + Z_k\}$                                   # Report noisy argmax

---

In the change-point or statistical process control (SPC) literature, when the pre- and post- change distributions are unknown in practical settings, researchers often choose hypotheses $P_0, P_1$ with the smallest justifiable distance. While it is easier to detect and accurately estimate a larger change, larger changes are often associated with a higher-sensitivity MLE, requiring more noise (and therefore additional error) to preserve privacy. We propose that practitioners using our private change point detection algorithm choose input hypotheses accordingly. This practical setting is considered in our numerical studies, presented in Section 5.

In the case that $\delta = 0$, we sample Laplace noise directly proportional to the sensitivity of the partial log-likelihood ratios we compute:

$$\Delta\ell \quad = \quad \max_{\substack{k \in [n], X, X' \in \mathbb{R}^n \\ ||X - X'||_1 = 1}} ||\ell(k, X) - \ell(k, X')||_1 = \max_{x \in \mathbb{R}} \log \frac{P_1(x)}{P_0(x)} - \min_{x' \in \mathbb{R}} \log \frac{P_1(x')}{P_0(x')}. \tag{4}$$

The algorithm should not be invoked with $\delta = 0$ unless $\Delta(\ell)$ is finite. In the case that $\ell$ has infinite sensitivity, we instead allow the user to select a privacy parameter $\delta > 0$ and identify a value $A_\delta$ for which most values of $x \sim P_0, P_1$ have bounded log-likelihood ratio:

$$A_\delta = \min\left\{t \ : \ \max_{i=0,1} \Pr_{x \sim P_i}\left[2|\log \frac{P_1(x)}{P_0(x)}| > t\right] < \delta/2\right\}. \tag{5}$$

As a concrete canonical example, $\Delta(\ell)$ is unbounded for two Gaussian distributions, but $A_\delta$ is bounded for Gaussians with different means as follows:

**Example 1.** *For $P_0 = \mathcal{N}(0, 1)$, $P_1 = \mathcal{N}(\mu, 1)$, and $\delta > 0$, we have $A_\delta = 2\mu[\Phi^{-1}(1 - \delta/2) + \mu/2]$, where $\Phi$ is the cumulative distribution function (CDF) of the standard normal distribution.*

### 3.2 Theoretical properties under the uniform bound assumption

In this subsection, we prove privacy and accuracy of OFFLINEPCPD when $\delta = 0$ and $P_0, P_1$ are such that $\Delta(\ell)$ is finite. Note that if $\Delta(\ell)$ is infinite, then the algorithm will simply add noise with infinite scale and will still be differentially private.

**Theorem 1.** *For arbitrary data $X$, OFFLINEPCPD$(X, P_0, P_1, \epsilon, 0)$ is $(\epsilon, 0)$-differentially private.*

The proof follows by instantiation of REPORTMAX [DR14] with queries $\ell(k)$ for $k \in [n]$, which have sensitivity $A = \Delta(\ell)$. It is included in Appendix C for completeness.

Next we provide accuracy guarantees of the standard (non-private) MLE $\hat{k}$ and the output $\tilde{k}$ of our private algorithm OFFLINEPCPD when the data are drawn from $P_0, P_1$ with true change point $k^* \in (1, n)$. By providing both bounds, Theorem 2 quantifies the cost of requiring privacy in change point detection.

Our result for the standard (non-private) MLE is the first finite-sample accuracy guarantee for this estimator. Such non-asymptotic properties have not been previously studied in traditional statistics, which typically focuses on consistency and unbiasedness of the estimator, with less attention to the convergence rate. We show that the additive error of the MLE is constant with respect to the sample size, which means that the convergence rate is $O_P(1)$. That is, it converges in probability to the true change-point $k^*$ in constant time.

Note that accuracy depends on two measures $A$ and $C$ of the distances between distributions $P_0$ and $P_1$. Accuracy both of MLE $\hat{k}$ and OFFLINEPCPD output $\tilde{k}$ is best for distributions for which $A = \Delta(\ell)$ is small relative to KL-divergence, which is consistent with the intuition that larger changes are easier to detect but output sensitivity degrades the robustness of the estimator and requires more noise for privacy, harming accuracy.

A technical challenge that arises in proving accuracy of the private estimator is that the $x_i$ are not identically distributed when the true change-point $k^* \in (1, n]$, and so the partial log-likelihood ratios $\ell(k)$ are dependent across $k$. Hence we need to investigate the impact of adding i.i.d. noise draws to a sequence of $\ell(k)$ that may be neither independent nor identically distributed. Fortunately, the differences $\ell(k) - \ell(k+1) = \log \frac{P_1(x_k)}{P_0(x_k)}$ are piecewise i.i.d. This property is key in our proof. Moreover, we show that we can divide the possible outputs of the algorithm into regions that of doubling size with exponentially decreasing probability of being selected by the algorithm, resulting in accuracy bounds that are independent of the number of data points $n$.

**Theorem 2.** *For hypotheses $P_0, P_1$ such that $\Delta(\ell) < \infty$ and $n$ data points $X$ drawn from $P_0, P_1$ with true change time $k^* \in (1, n]$, the MLE $\hat{k}$ is $(\alpha, \beta)$-accurate for any $\beta > 0$ and*

$$\alpha = \frac{2A^2}{C^2} \log \frac{32}{3\beta}. \tag{6}$$

*For hypotheses and data drawn this way with privacy parameter $\epsilon > 0$, OFFLINEPCPD$(X, P_0, P_1, \epsilon, 0, n)$ is $(\alpha, \beta)$-accurate for any $\beta > 0$ and*

$$\alpha = \max \left\{ \frac{8A^2}{C^2} \log \frac{64}{3\beta}, \frac{4A}{C\epsilon} \log \frac{16}{\beta} \right\}. \tag{7}$$

*In both expressions, $A = \Delta(\ell)$ and $C = \min\{D_{KL}(P_1||P_0), D_{KL}(P_0||P_1)\}$.*

### 3.3 Relaxing uniform bound assumptions

In this subsection, we prove accuracy and a limited notion of privacy for OFFLINEPCPD when $\delta > 0$ and $P_0, P_1$ are such that $A_\delta$ is finite. Since we are no longer able to uniformly bound $\log P_1(x)/P_0(x)$, these accuracy results include worse constants than those in Section 3.2, but the relaxed assumption about $P_0, P_1$ makes the results applicable to a wider range of distributions, including Gaussian distributions (see Example 1). Note of course that for some pairs of very different distributions, such as distributions with non-overlapping supports, the assumption that $A_\delta < \infty$ may still fail. A true change point $k^*$ can always be detected with perfect accuracy given $x_{k^*-1}$ and $x_{k^*}$, so we should not expect to be able to offer any meaningful privacy guarantees for such distributions.

By similar rationale, relaxing the uniform bound assumption means that we may have a single data point $x_j$ that dramatically increases $\ell(k)$ for $k \geq j$, so we cannot add noise proportional to $\Delta(\ell)$ and privacy no longer follows from that of REPORTMAX. Instead we offer a weaker notion of privacy in Theorem 3 below. As with the usual definition of differential privacy, we guarantee that the output of our algorithm is similarly distributed on neighboring databases, only our notion of neighboring databases depends on the hypothesized distributions. Specifically, the a single entry in $X$ drawn from either $P_0$ or $P_1$ may be replaced without detection by another entry drawn from either $P_0$ or $P_1$, even if the rest of the database is arbitrary. The proof is given in Appendix C.

**Theorem 3.** *For any $\epsilon, \delta > 0$, any hypotheses $P_0, P_1$ such that $A_\delta < \infty$, any index $j \in [n]$, any $i, i' \in \{0, 1\}$, and any $x_1, \ldots, x_{j-1}, x_{j+2}, \ldots, x_n$, let $X_i = \{x_1, \ldots, x_n\}$ denote the random*

*variable with $x_j \sim P_i$ and let $X'_{i'} = \{x_1, \ldots, x_{j-1}, x'_j, x_{j+1}, \ldots, x_n\}$ denote the random variable with $x'_j \sim P_{i'}$. Then for any $S \subseteq [n]$, we have*

$$\Pr[\text{OFFLINEPCPD}(X_i, P_0, P_1, \epsilon, \delta, n) \in S]$$
$$\leq \exp(\epsilon) \cdot \Pr[\text{OFFLINEPCPD}(X'_{i'}, P_0, P_1, \epsilon, \delta, n) \in S] + \delta,$$

*where the probabilities are over the randomness of the algorithm and of $X_i, X'_{i'}$.*

Allowing $\Delta(\ell)$ to be infinite precludes our use of Hoeffding's inequality as in Theorem 2. The main idea in the proof, however, can be salvaged by decomposing the change into a change from $P_0$ to the average distribution $(P_0 + P_1)/2$ and then the average distribution to $P_1$. Correspondingly, we will use $C_M$, an alternate distance measure between $P_0$ and $P_1$, defined below next to $C$ from the previous section for comparison:

$$C = \min\{D_{KL}(P_0||P_1), D_{KL}(P_1||P_0)\} \tag{8}$$

$$C_M = \min\left\{D_{KL}(P_0||\frac{P_0 + P_1}{2}), D_{KL}(P_1||\frac{P_0 + P_1}{2})\right\} = \min_{i=0,1} \mathbb{E}_{x \sim P_i}\left[\log\frac{2P_i(x)}{P_0(x) + P_1(x)}\right] \tag{9}$$

Because $(2P_i)/(P_0 + P_1) \leq 2$, we have $0 \leq D_{KL}(P_i||(P_0 + P_1)/2) \leq \log 2$, and thus the constant $C_M$ in (9) is well-defined. The proof of the following theorem is given in Appendix C.

**Theorem 4.** *For $\delta > 0$ and hypotheses $P_0, P_1$ such that $A_\delta < \infty$ and $n$ data points $X$ drawn from $P_0, P_1$ with true change time $k^* \in (1, n)$, the MLE $\hat{k}$ is $(\alpha, \beta)$-accurate for any $\beta > 0$ and*

$$\alpha = \frac{67}{C_M^2}\log\frac{64}{3\beta}. \tag{10}$$

*For hypotheses and data drawn this way with privacy parameter $\epsilon > 0$, $\text{OFFLINEPCPD}(X, P_0, P_1, \epsilon, \delta, n)$ is $(\alpha, \beta)$-accurate for any $\beta > 0$ and*

$$\alpha = \max\left\{\frac{262}{C_M^2}\log\frac{128}{3\beta}, \frac{2A\log(16/\beta)}{C_M\epsilon}\right\}. \tag{11}$$

*In both expressions, $A = A_\delta$ and $C_M = \min\left\{D_{KL}(P_0||\frac{P_0 + P_1}{2}), D_{KL}(P_1||\frac{P_0 + P_1}{2})\right\}$.*

## 4 Online private change-point detection

In this section, we give a new differentially private algorithm for change point detection in the online setting, ONLINEPCPD. In this setting, the algorithm initially receives $n$ data points $x_1, \ldots, x_n$ and then continues to receive data points one at a time. As before, the goal is to privately identify an approximation of the time $k^*$ when the data change from distribution $P_0$ to $P_1$. Additionally, we want to identify this change shortly after it occurs.

Our offline algorithm is not directly applicable because we do not know a priori how many points must arrive before a true change point occurs. To resolve this, ONLINEPCPD works like ABOVETHRESH, determining after each new data entry arrives whether it is likely that a change occurred in the most recent $n$ entries. When ONLINEPCPD detects a sufficiently large (noisy) partial log likelihood ratio $\ell(k) = \sum_{i=k}^{j}\log\frac{P_1(x_i)}{P_0(x_i)}$, it calls OFFLINEPCPD to privately determine the most likely change point $\tilde{k}$ in the window $\{x_{j-n+1}, \ldots, x_j\}$.

Privacy of ONLINEPCPD is immediate from composition of ABOVETHRESH and OFFLINEPCPD, each with privacy loss $\epsilon/2$. As before, accuracy requires $X$ to be drawn from $P_0, P_1$ with some true change point $k^*$. This algorithm also requires a suitable choice of $T$ to guarantee that OFFLINEPCPD is called for a window of data that actually contains $k^*$. Specifically, $T$ should be large enough that the algorithm is unlikely to call OFFLINEPCPD when $j < k^*$ but small enough so that it is likely to call OFFLINEPCPD by time $j = k^* + n/2$. When both of these conditions hold, we inherit the accuracy of OFFLINEPCPD, with an extra $\log n$ factor arising from the fact that the data are no longer distributed exactly as in the change-point model after conditioning on calling OFFLINEPCPD in a correct window.

With our final bounds, we note that $n \gg \frac{A}{C} \log(k^*/\beta)$ suffices for existence of a suitable threshold, and an analyst must have a reasonable approximation of $k^*$ in order to choose such a threshold. Otherwise, the accuracy bound itself has no dependence on the change-point $k^*$.

---

**Algorithm 2** Online private change-point detector : $\text{ONLINEPCPD}(X, P_0, P_1, \epsilon, n, T)$

---

**Input:** database $X$, distributions $P_0, P_1$, privacy parameter $\epsilon$, starting size $n$, threshold $T$

Let $A = \max_x \log \frac{P_1(x)}{P_0(x)} - \min_{x'} \log \frac{P_1(x')}{P_0(x')}$

Let $\hat{T} = T + \text{Lap}(4A/\epsilon)$

**for** each new data point $x_j, j \geq n$ **do**

    Compute $\ell_j = \max_{j-n+1 \leq k \leq j} \ell(k)$

    Sample $Z_j \sim \text{Lap}(\frac{8A}{\epsilon})$

    **if** $\ell_j + Z_j > \hat{T}$ **then**

        Output $\text{OFFLINEPCPD}(\{x_{j-n+1}, \ldots, x_j\}, P_0, P_1, \epsilon/2, 0, n) + (j - n)$

        Halt

    **else**

        Output $\perp$

    **end if**

**end for**

---

**Theorem 5.** *For arbitrary data $X$, $\text{ONLINEPCPD}(X, P_0, P_1, \epsilon, n, T)$ is $(\epsilon, 0)$-differentially private.*

This privacy guarantee follows from simple composition of ABOVETHRESH and OFFLINEPCPD, each with privacy loss $\epsilon/2$. The proof of the accuracy bound is given in Appendix D.

**Theorem 6.** *For hypotheses $P_0, P_1$ such that $\Delta(\ell) < \infty$, a stream of data points $X$ with starting size $n$ drawn from $P_0, P_1$ with true change time $k^* \geq n/2$, privacy parameter $\epsilon > 0$, and threshold $T \in [T_L, T_U]$ with*

$$T_L \quad := \quad 2A\sqrt{2 \log \frac{64k^*}{\beta}} - C + \frac{16A}{\epsilon} \log \frac{8k^*}{\beta},$$

$$T_U \quad := \quad \frac{nC}{2} - \frac{A}{2}\sqrt{n \log(8/\beta)} - \frac{16A}{\epsilon} \log \frac{8k^*}{\beta},$$

*we have that $\text{ONLINEPCPD}(X, P_0, P_1, \epsilon, n, T)$ is $(\alpha, \beta)$ accurate for any $\beta > 0$ and*

$$\alpha = \max \left\{ \frac{16A^2}{C^2} \log \frac{32n}{\beta}, \frac{4A}{C\epsilon} \log \frac{8n}{\beta} \right\}.$$

*In the above expressions, $A = \Delta(\ell)$ and $C = \min\{D_{KL}(P_0||P_1), D_{KL}(P_1||P_0)\}$.*

## 5 Numerical studies

We now report the results of Monte Carlo experiments designed to validate the theoretical results of previous sections. We only consider our accuracy guarantees because the nature of differential privacy provides a strong worst-case guarantee for all hypothetical databases, and therefore is impractical and redundant to test empirically. Our simulations consider both offline and online settings for two canonical problems: detecting a change in the mean of Bernoulli and Gaussian distributions.

We begin with the offline setting to verify performance of our OFFLINEPCPD algorithm. We use $n = 200$ observations where the true change occurs at time $k^* = 100$. This process is repeated $10^4$ times. For both the Bernoulli and Gaussian models, we consider the following three different change scenarios, corresponding to the size of the change and parameter selection for OFFLINEPCPD. For each of these cases, we consider privacy parameter $\epsilon = 0.1, 0.5, 1, \infty$, where $\epsilon = \infty$ corresponds to the non-private problem, which serves as our baseline. The results are summarized in Figure 1, which plots the empirical probabilities $\beta = \Pr[|\tilde{k} - k^*| > \alpha]$ as a function of $\alpha$.

    (A) **Large change.** Bernoulli model: detecting a change from $p_0 = 0.2$ to $p_1 = 0.8$. Gaussian model: detecting a change from $\mu_0 = 0$ to $\mu_1 = 1$.

(B) **Small change.** Bernoulli model: detecting a change from $p_0 = 0.2$ to $p_1 = 0.4$. Gaussian model: detecting a change from $\mu_0 = 0$ to $\mu_1 = 0.5$.

(C) **Misspecified change** Bernoulli model: algorithm tests for change from $p_0 = 0.2$ to $p_1 = 0.4$ when true distributions have $p_0 = 0.2$ and $p_1 = 0.8$. Gaussian model: algorithm tests for change from $\mu_0 = 0$ to $\mu_1 = 0.5$ when true distributions have $\mu_0 = 0$ and $\mu_1 = 1$.

Figure 1 highlights three positive results for our algorithm when data is drawn from Bernoulli or Gaussian distributions: accuracy is best when the true change in data is large (plots a and d) compared to small (plots b and e), accuracy deteriorates as $\epsilon$ decreases for stronger privacy, and the algorithm performs well even when the true change is larger than that hypothesized (plots c and f). This figure emphasizes that our algorithm performs well even for quite strong privacy guarantees ($\epsilon < 1$). The misspecified change experiments bolster our theoretical results substantially, indicating that our hypotheses can be quite far from the distributions of the true data and our algorithms will still identify a change-point accurately. We also run Monte Carlo simulations of our online change-point detection algorithm ONLINEPCPD. These are displayed in Figure 2 and discussed in Appendix E.

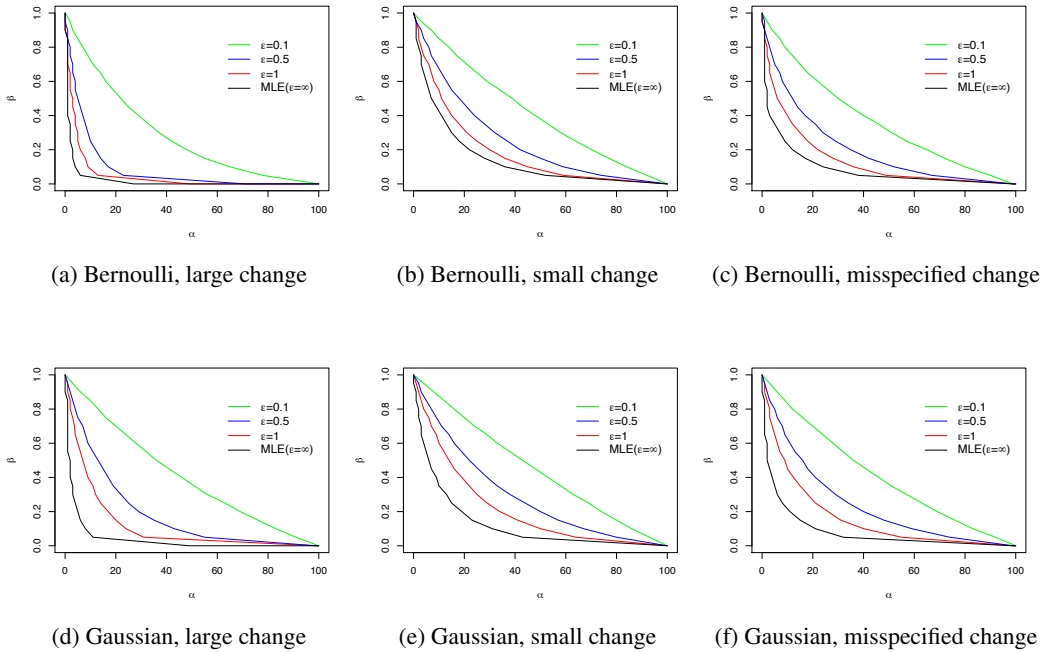

(a) Bernoulli, large change     (b) Bernoulli, small change     (c) Bernoulli, misspecified change

(d) Gaussian, large change     (e) Gaussian, small change     (f) Gaussian, misspecified change

Figure 1: Accuracy for large change, small change, and misspecified change Monte Carlo simulations with Bernoulli and Gaussian data. Each simulation involves $10^4$ runs of OFFLINEPCPD with varying $\epsilon$ on data generated by 200 i.i.d. samples from appropriate distributions with change point $k^* = 100$.

## Acknowledgments

R.C. and S.K. were supported in part by a Mozilla Research Grant. Y.M. and W.Z. were supported in part by NSF grant CMMI-1362876. R.T. was supported in part by NSF grant DMS-156443. R.T.'s contribution was completed while the author was visiting the Georgia Institute of Technology.

## Footnotes

[2]We note that we can relax our distributional assumption and get a weaker privacy guarantee as in the offline setting if desired.

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
