[Supplementary Material · neurips_2018_SupplementaryMaterial.pdf]

# A Differential privacy background

The REPORTMAX algorithm takes in a collection of queries, computes a noisy answer to each query, and returns the index of the query with the largest noisy value. We use this as the framework for our offline private change-point detector OFFLINEPCPD in Section 3 to privately select the time $k$ with the highest log-likelihood ratio $\ell(k)$.

---

**Algorithm 3** Report Noisy Max : REPORTMAX($X, \Delta, \{f_1, \ldots, f_n\}, \epsilon$)

---

    **Input:** database $X$, set of queries $\{f_1, \ldots, f_n\}$ each with sensitivity $\Delta$, and privacy parameter $\epsilon$
    **for** $i = 1, \ldots, n$ **do**
        Compute $f_i(X)$
        Sample $Z_i \sim \text{Lap}(\frac{\Delta}{\epsilon})$
    **end for**
    Output $i^* = \underset{i \in [n]}{\text{argmax}} \, (f_i(X) + Z_i)$

---

**Theorem 7** ([DR14]). REPORTMAX *is* $(\epsilon, 0)$-*differentially private.*

The ABOVETHRESH algorithm, first introduced by [DNR$^+$09] and refined to its current form by [DR14], takes in a potentially unbounded stream of queries, compares the answer of each query to a fixed noisy threshold, and halts when it finds a noisy answer that exceeds the noisy threshold. We use this algorithm as a framework for our online private change-point detector ONLINEPCPD in Section 4 when new data points arrive online in a streaming fashion.

---

**Algorithm 4** Above Noisy Threshold: ABOVETHRESH($X, \Delta, \{f_1, f_2, \ldots\}, T, \epsilon$)

---

    **Input:** database $X$, stream of queries $\{f_1, f_2, \ldots\}$ each with sensitivity $\Delta$, threshold $T$, and privacy parameter $\epsilon$
    Let $\hat{T} = T + \text{Lap}(\frac{2\Delta}{\epsilon})$
    **for** Each query $i$ **do**
        Let $Z_i \sim \text{Lap}(\frac{4\Delta}{\epsilon})$
        **if** $f_i(X) + Z_i > \hat{T}$ **then**
            Output $a_i = \top$.
            Halt.
        **else**
            Output $a_i = \bot$
        **end if**
    **end for**

---

**Theorem 8** ([DNR$^+$09]). ABOVETHRESH *is* $(\epsilon, 0)$-*differentially private.*

**Theorem 9** ([DNR$^+$09]). *For any sequence of $n$ queries $f_1, \ldots, f_n$ such that $|\{i < n : f_i(X) \geq T - \alpha\}| = 0$, Above Noisy Threshold outputs with probability at least $1 - \beta$ a stream of $a_1, \ldots, a_n \in \{\top, \bot\}$ such that for all $a_i = \top$, $f_i(X) \geq T - \alpha$, and for all $a_i = \bot$, $f_i(X) \leq T + \alpha$, for*

$$\alpha = \frac{8(\log k + \log(2/\beta))}{\epsilon}.$$

# B Concentration inequalities

Our proofs will use the following bounds.

**Lemma 1** (Ottaviani's inequality [VDVW96]). *For independent random variables $U_1, \ldots, U_m$, for $S_k = \sum_{i \in [k]} U_i$ for $k \in [m]$, and for $\lambda_1, \lambda_2 > 0$, we have*

$$\Pr\left[\max_{1 \leq k \leq m} |S_k| > \lambda_1 + \lambda_2\right] \leq \frac{\Pr\left[|S_m| > \lambda_1\right]}{1 - \max_{1 \leq k \leq m} \Pr\left[|S_m - S_k| > \lambda_2\right]}.$$

If we additionally assume the $U_j$ above are i.i.d. with mean 0 and take values from an interval of bounded length $L$, we can apply Hoeffding's inequality for the following corollary:

**Corollary 1.** *For independent and identically distributed random variables $U_1, \ldots, U_m$ with mean zero strictly bounded by an interval of length $L$ and for $S_k = \sum_{i \in [k]} U_i$ for $k \in [m]$, and for $\lambda_1, \lambda_2 > 0$, we have*

$$\Pr[\max_{k \in [m]} |S_k| > \lambda_1 + \lambda_2] \leq \frac{2 \exp(-2\lambda_1^2/(mL^2))}{1 - 2\exp(-2\lambda_2^2/(mL^2))}.$$

**Lemma 2** (Bernstein inequality [VDVW96])**.** *Let $Y_1, \ldots, Y_n$ be independent random variables with mean zero such that $\mathbb{E}\left[e^{|Y_i|/M} - 1 - \frac{|Y_i|}{M}\right] M^2 \leq \frac{1}{2} v_i$ for constants $M$ and $v_i$ and for $i \in [n]$. Then*

$$\Pr[|Y_1 + \ldots + Y_n| > x] \leq 2\exp\left(-\frac{1}{2}\frac{x^2}{v + Mx}\right),$$

*for $v \geq v_1 + \ldots + v_n$.*

**Corollary 2.** *For independent and identically distributed random variables $Y_1, \ldots, Y_n$ with mean zero such that $\mathbb{E}\left[e^{|Y_i|} - 1 - |Y_i|\right] \leq \frac{1}{2} v$, for constant $v$ and $i \in [n]$, and for $S_k = \sum_{i \in [k]} Y_i$ for $k \in [m]$, and for $\lambda_1, \lambda_2 > 0$, we have*

$$\Pr[\max_{k \in [m]} |S_k| > \lambda_1 + \lambda_2] \leq \frac{2 \exp(-\lambda_1^2/(2mv + 2\lambda_1))}{1 - 2\exp(-\lambda_2^2/(2mv + 2\lambda_2))}.$$

## C  Proofs for OFFLINEPCPD

**Theorem 1.** *For arbitrary data $X$, OFFLINEPCPD$(X, P_0, P_1, \epsilon, 0)$ is $(\epsilon, 0)$-differentially private.*

*Proof.* Fix any two neighboring databases $X, X'$ that differ on index $j$. For any $k \in [n]$, denote the respective partial log-likelihood ratios as $\ell(k)$ and $\ell'(k)$. By (1), we have

$$\ell'(k) = \ell(k) + \Delta \mathbb{I}\{j \geq k\} \qquad \text{with} \quad \Delta = \log\frac{P_1(x'_j)}{P_0(x'_j)} - \log\frac{P_1(x_j)}{P_0(x_j)}. \tag{12}$$

Next, for a given $1 \leq i \leq n$, fix $Z_{-i}$, a draw from $[\text{Lap}(A/\epsilon)]^{n-1}$ used for all the noisy log likelihood ratio values except the $i$th one. We will bound from above and below the ratio of the probabilities that the algorithm outputs $\tilde{k} = i$ on inputs $X$ and $X'$. Define the minimum noisy value in order for $i$ to be select with $X$:

$$Z_i^* = \min\{Z_i : \ell(i) + Z_i > \ell(k) + Z_k \quad \forall k \neq i\}$$

If $\Delta < 0$, then for all $k \neq i$ we have

$$\ell'(i) + A + Z_i^* \geq \ell(i) + Z_i^* > \ell(k) + Z_k \geq \ell'(k) + Z_k.$$

If $\Delta \geq 0$, then for all $k \neq i$ we have

$$\ell'(i) + Z_i^* \geq \ell(i) + Z_i^* > \ell(k) + Z_k \geq \ell'(k) - A + Z_k.$$

Hence, $Z_i' \geq Z_i^* + A$ ensures that the algorithm outputs $i$ on input $X'$, and the theorem follows from the following inequalities for any fixed $Z_{-i}$, with probabilities over the choice of $Z_i \sim \text{Lap}(A/\epsilon)$.

$$\Pr[\tilde{k} = i \mid X', Z_{-i}] \geq \Pr[Z_i' \geq Z_i^* + A \mid Z_{-i}] \geq e^{-\epsilon}\Pr[Z_i \geq Z_i^* \mid Z_{-i}] = e^{-\epsilon}\Pr[\tilde{k} = i \mid X, Z_{-i}]$$
$$\square$$

**Theorem 2.** *For hypotheses $P_0, P_1$ such that $\Delta(\ell) < \infty$ and $n$ data points $X$ drawn from $P_0, P_1$ with true change time $k^* \in (1, n]$, the MLE $\hat{k}$ is $(\alpha, \beta)$-accurate for any $\beta > 0$ and*

$$\alpha = \frac{2A^2}{C^2}\log\frac{32}{3\beta}. \tag{6}$$

*For hypotheses and data drawn this way with privacy parameter $\epsilon > 0$, OFFLINEPCPD$(X, P_0, P_1, \epsilon, 0, n)$ is $(\alpha, \beta)$-accurate for any $\beta > 0$ and*

$$\alpha = \max\left\{\frac{8A^2}{C^2}\log\frac{64}{3\beta}, \frac{4A}{C\epsilon}\log\frac{16}{\beta}\right\}. \tag{7}$$

*In both expressions, $A = \Delta(\ell)$ and $C = \min\{D_{KL}(P_1||P_0), D_{KL}(P_0||P_1)\}$.*

*Proof.* Our goal is to find some expression for $\alpha$ such that we can bound the probability of the bad event that OFFLINEPCPD outputs $\tilde{k}$ such that $|\tilde{k} - k^*| > \alpha$ with probability at most $\beta$, where $k^*$ is the true change point. The first half of our analysis will yield another bound giving accuracy of the MLE $\hat{k}$.

Our proof is structured around the following observation. The algorithm only outputs a particular incorrect $\tilde{k} \neq k^*$ if there exists some $k$ in with $\ell(k) + Z_k > \ell(k^*) + Z_{k^*}$ for a set of random noise values $\{Z_k\}_{k \in [n]}$ selected by the algorithm. For the algorithm to output an incorrect value, there must either be a $k$ that nearly beats the true change point on the noiseless data or there must be a $k$ that receives much more noise than $k^*$. Intuitively, this captures the respective scenarios that unusual data causes non-private ERM to perform poorly and that unusual noise draws causes our private algorithm to perform poorly.

Given some true change-point $k^*$ and error tolerance $\alpha > 0$, we can partition the set of bad possible outputs $k$ into sub-intervals of exponentially increasing size as follows. For $i \geq 1$, let

$$
\begin{aligned}
R_i^- &= [k^* - 2^i\alpha, k^* - 2^{i-1}\alpha) \\
R_i^+ &= (k^* + 2^{i-1}\alpha, k^* + 2^i\alpha] \\
R_i &= R_i^- \cup R_i^+
\end{aligned}
$$

Then for any range-specific thresholds $t_i$ for $i \geq 1$, our previous observations allow us to bound the probability of the bad event as follows:

$$
\Pr[|\tilde{k} - k^*| > \alpha] \leq \sum_{i \geq 1} \Pr[\max_{k \in R_i}\{\ell(k) - \ell(k^*)\} > -t_i] + \sum_{i \geq 1} \Pr[\max_{k \in R_i}\{Z_k - Z_{k^*}\} \geq t_i] \quad (13)
$$

We bound each term in the above expression separately for $t_i = 2^{i-2}\alpha C$. For accuracy of the non-private MLE, we will set $\alpha$ to ensure that the first term is at most $\beta$. For accuracy of the private algorithm, we will set $\alpha$ to ensure that each term is at most $\beta/2$. The first and more difficult task requires us to reason about the probability that the log-likelihood ratios for the data are not too far away from their expectation. Although the $\ell(k)$ are not independent, their pairwise differences $\ell(k+1) - \ell(k)$ are, so we can apply our corollary of Ottaviani's inequality to bound the probability that $\ell(k)$ significantly exceeds $\ell(k^*)$ by appropriately defining several random variables corresponding to a data stream $X$ drawn according to the change-point model.

Specifically, we can decompose the empirical log-likelihood difference between the true change-point $k^*$ and any candidate $k$ into the sum of i.i.d. random variables with mean zero and the expected value of this difference as follows:

$$
\begin{aligned}
U_j &= \begin{cases} -\log \frac{P_0(x_j)}{P_1(x_j)} + D_{KL}(P_0 \| P_1), & j < k^* \\ -\log \frac{P_1(x_j)}{P_0(x_j)} + D_{KL}(P_1 \| P_0), & j \geq k^* \end{cases} \\
\ell(k) - \ell(k^*) &= \begin{cases} \sum_{j=k}^{k^*-1} U_j - (k^* - k)D_{KL}(P_0 \| P_1), & k < k^* \\ \sum_{j=k^*}^{k-1} U_j - (k - k^*)D_{KL}(P_1 \| P_0), & k \geq k^* \end{cases}
\end{aligned}
$$

We also define random variable $S_m$ to denote the sum of $m$ i.i.d. random variables as follows, noting that $S_m$ is distributed like $\sum_{j=k^*+m}^{k^*-1} U_j$ for $m < 0$ and like $\sum_{j=k^*}^{k^*+m-1} U_j$ for $m > 0$.

$$
S_m = \begin{cases} \sum_{k^*+m \leq j < k^*} U_j, & m < 0 \\ \sum_{k^* \leq j < k^*+m} U_j & m > 0 \end{cases}
$$

With these random variables, we bound each term in the first set of terms in (13) for any $i \geq 1$ and threshold $t_i = 2^{i-2}\alpha C$ as follows:

$$\Pr[\max_{k \in R_i}\{\ell(k) - \ell(k^*)\} > -2^{i-2}\alpha C]$$

$$\leq \Pr[\max_{k \in R_i^-}\{\sum_{j=k}^{k^*-1} U_j - (k^* - k)D_{KL}(P_0||P_1)\} > -2^{i-2}\alpha C]$$

$$+ \Pr[\max_{k \in R_i^+}\{\sum_{j=k^*}^{k-1} U_j - (k - k^*)D_{KL}(P_1||P_0)\} > -2^{i-2}\alpha C]$$

$$\leq \Pr[\max_{k \in [2^{i-1}\alpha]}|S_{-k}| > 2^{i-2}\alpha C] + \Pr[\max_{k \in [2^{i-1}\alpha]}|S_k| > 2^{i-2}\alpha C]$$

$$\leq \frac{4 \cdot \exp(-2^{i-4}\alpha C^2/A^2)}{1 - 2 \cdot \exp(-2^{i-4}\alpha C^2/A^2)} \tag{14}$$

$$\leq 8\exp(-2^{i-4}\alpha C^2/A^2) \tag{15}$$

$$= 8\left(\exp(\frac{-\alpha C^2}{8A^2})\right)^{2^{i-1}}$$

where (17) follows from an application of Corollary 1 with $\lambda_1 = \lambda_2 = 2^{i-3}\alpha C$ and $L = A$, and the denominator can be simplified as in (18) under the assumption that $\alpha \geq \frac{8A^2 \log 4}{C^2}$ to simplify the denominator, which is satisfied by our final bounds.

We now consider the sum of these terms over all $i$, which will be needed for the final bound on Equation (13). We note that this sum is bounded above by a geometric series with ratio $\exp(-\alpha C^2/(8A^2))$ since $2^{i-1} \geq i$, yielding the second inequality. Then the same assumed lower bound on $\alpha$ is used to simplify the denominator as in (18):

$$\sum_{i \geq 1}\Pr[\max_{k \in R_i}\{\ell(k) - \ell(k^*)\} > -2^{i-2}\alpha C] \leq 8\sum_{i \geq 1}\left(\exp(\frac{-\alpha C^2}{8A^2})\right)^{2^{i-1}}$$

$$\leq 8\sum_{i \geq 1}\left(\exp(\frac{-\alpha C^2}{8A^2})\right)^{i}$$

$$\leq \frac{8\exp(\frac{-\alpha C^2}{8A^2})}{1 - \exp(\frac{-\alpha C^2}{8A^2})}$$

$$\leq \frac{32}{3}\exp\left(\frac{-\alpha C^2}{8A^2}\right) \tag{16}$$

The first term in (7) in the theorem statement ensures that the expression above is bounded by $\beta/2$, as is required for the private algorithm.

For non-private MLE, we bound each term in the first set of terms in (13) for any $i \geq 1$ and threshold $t_i = 0$ as follows:

$$\Pr[\max_{k \in R_i}\{\ell(k) - \ell(k^*)\} > 0]$$

$$\leq \Pr[\max_{k \in R_i^-}\{\sum_{j=k}^{k^*-1} U_j - (k^* - k)D_{KL}(P_0||P_1)\} > 0]$$

$$+ \Pr[\max_{k \in R_i^+}\{\sum_{j=k^*}^{k-1} U_j - (k - k^*)D_{KL}(P_1||P_0)\} > 0]$$

$$\leq \Pr[\max_{k \in [2^{i-1}\alpha]} |S_{-k}| > 2^{i-1}\alpha C] + \Pr[\max_{k \in [2^{i-1}\alpha]} |S_k| > 2^{i-1}\alpha C]$$

$$\leq \frac{4 \cdot \exp(-2^{i-2}\alpha C^2/A^2)}{1 - 2 \cdot \exp(-2^{i-2}\alpha C^2/A^2)} \tag{17}$$

$$\leq 8 \exp(-2^{i-2}\alpha C^2/A^2) \tag{18}$$

$$= 8 \left( \exp(\frac{-\alpha C^2}{2A^2}) \right)^{2^{i-1}}$$

Summing these terms over all $i$,

$$\sum_{i \geq 1} \Pr[\max_{k \in R_i}\{\ell(k) - \ell(k^*)\} > 0] \leq 8 \sum_{i \geq 1} \left( \exp(\frac{-\alpha C^2}{2A^2}) \right)^{2^{i-1}}$$

$$\leq 8 \sum_{i \geq 1} \left( \exp(\frac{-\alpha C^2}{2A^2}) \right)^{i}$$

$$\leq \frac{8 \exp(\frac{-\alpha C^2}{2A^2})}{1 - \exp(\frac{-\alpha C^2}{2A^2})}$$

$$\leq \frac{32}{3} \exp\left( \frac{-\alpha C^2}{2A^2} \right). \tag{19}$$

For $\alpha$ as in (6) in the theorem statement, the expression above is bounded by $\beta$, completing the accuracy proof for the non-private MLE.

Next we bound the second set of terms in (13), controlling the probability that large noise draws cause large inaccuracies for the private algorithm. Since each $Z_k$ and $Z_{k^*}$ are independent draws from a Laplace distribution with parameter $A/\epsilon$, this bound follows from a union bound over all indices in $R_i$ and the definition of the Laplace distribution:

$$\Pr[\max_{k \in R_i}\{Z_k - Z_{k^*}\} \geq 2^{i-2}\alpha C] \leq \Pr[2 \max_{k \in R_i} |Z_k| \geq 2^{i-2}\alpha C]$$

$$\leq 2^i \alpha \Pr[|\text{Lap}(A/\epsilon)| \geq 2^{i-3}\alpha C]$$

$$\leq 2^i \alpha \cdot \exp(-2^{i-3}\alpha C\epsilon/A)$$

$$= 2^i \alpha \left( \exp(\frac{-\alpha C\epsilon}{4A}) \right)^{2^{i-1}}$$

Then by summing over all ranges and assuming in (20) that $\alpha \geq \frac{4A \ln 2}{C\epsilon}$ to simplify the denominator, we obtain a bound on the probability of large noise applied to any possible $k$ far from $k^*$.

$$\sum_{i \geq 1} \Pr[\max_{k \in R_i}\{Z_k - Z_{k^*}\} > 2^{i-2}\alpha C] \leq \alpha \sum_{i \geq 1} 2^i (\exp(-\alpha C \epsilon / (4A)))^{2^{i-1}}$$

$$\leq \alpha 2 \sum_{i \geq 1} i (\exp(-\alpha C \epsilon / (4A)))^i$$

$$= \alpha 2 \frac{\exp(-\alpha C \epsilon / (4A))}{(1 - \exp(-\alpha C \epsilon / (4A)))^2}$$

$$\leq 8\alpha \exp(-\alpha C \epsilon / (4A)) \tag{20}$$

Since $x/2 \geq \ln x$, requiring $\alpha \geq \frac{4A \log(16/\beta)}{C\epsilon}$ suffices to ensure that (20) is at most $\beta/2$ as required. By Inequality 13, this guarantees that $\Pr[|\tilde{k} - k^*| > \alpha] \leq \beta$ for the assumed ranges of $\alpha$ captured in Equation (7) in the theorem statement, completing the proof.

$\square$

**Theorem 3.** *For any $\epsilon, \delta > 0$, any hypotheses $P_0, P_1$ such that $A_\delta < \infty$, any index $j \in [n]$, any $i, i' \in \{0, 1\}$, and any $x_1, \ldots, x_{j-1}, x_{j+2}, \ldots, x_n$, let $X_i = \{x_1, \ldots, x_n\}$ denote the random variable with $x_j \sim P_i$ and let $X'_{i'} = \{x_1, \ldots, x_{j-1}, x'_j, x_{j+1}, \ldots, x_n\}$ denote the random variable with $x'_j \sim P_{i'}$. Then for any $S \subseteq [n]$, we have*

$$\Pr[\text{OFFLINEPCPD}(X_i, P_0, P_1, \epsilon, \delta, n) \in S]$$
$$\leq \exp(\epsilon) \cdot \Pr[\text{OFFLINEPCPD}(X'_{i'}, P_0, P_1, \epsilon, \delta, n) \in S] + \delta,$$

*where the probabilities are over the randomness of the algorithm and of $X_i, X'_{i'}$.*

*Proof.* Define the event that the log-likelihood ratios of $x_j, x'_j$ as in the theorem statement are bounded by $A_\delta$ as follows:

$$E_\delta := \left\{ \left| \log \frac{P_1(x_j)}{P_0(x_j)} - \log \frac{P_1(x'_j)}{P_0(x'_j)} \right| < A_\delta \right\}.$$

Let $\tilde{k} = \text{OFFLINEPCPD}(X_i, P_0, P_1, \epsilon, \delta, n)$, $\tilde{k}' = \text{OFFLINEPCPD}(X'_{i'}, P_0, P_1, \epsilon, \delta, n)$. Then by Theorem 1 and the observation that $\Pr[E_\delta^c] < \delta$ by definition of $A_\delta$, we have that for any $S \subseteq [n]$,

$$\begin{aligned} \Pr[\tilde{k} \in S] &\leq& \Pr[\tilde{k} \in S | E_\delta] \Pr[E_\delta] + \Pr[E_\delta^c] \\ &\leq& \exp(\epsilon) \Pr[\tilde{k}' \in S | E_\delta] \Pr[E_\delta] + \delta \\ &\leq& \exp(\epsilon) \Pr[\tilde{k}' \in S] + \delta. \end{aligned}$$

$\square$

**Theorem 4.** *For $\delta > 0$ and hypotheses $P_0, P_1$ such that $A_\delta < \infty$ and $n$ data points $X$ drawn from $P_0, P_1$ with true change time $k^* \in (1, n)$, the MLE $\hat{k}$ is $(\alpha, \beta)$-accurate for any $\beta > 0$ and*

$$\alpha = \frac{67}{C_M^2} \log \frac{64}{3\beta}. \tag{10}$$

*For hypotheses and data drawn this way with privacy parameter $\epsilon > 0$, $\text{OFFLINEPCPD}(X, P_0, P_1, \epsilon, \delta, n)$ is $(\alpha, \beta)$-accurate for any $\beta > 0$ and*

$$\alpha = \max\{\frac{262}{C_M^2} \log \frac{128}{3\beta}, \frac{2A \log(16/\beta)}{C_M \epsilon}\}. \tag{11}$$

*In both expressions, $A = A_\delta$ and $C_M = \min\left\{D_{KL}(P_0 || \frac{P_0 + P_1}{2}), D_{KL}(P_1 || \frac{P_0 + P_1}{2})\right\}$.*

*Proof.* The general framework of this proof is similar to that of Theorem 2, but the main difference is that Hoeffding's inequality is not applicable in this general setting, since we allow $\Delta(\ell)$ to be unbounded. The main idea in this proof is to consider the alternative log-likelihood ratio using the average distribution $(P_0 + P_1)/2$, in which Bernstein inequality can be applied.

Following the notation from Theorem 2, given some true change-point $k^*$ and error tolerance $\alpha > 0$, we can partition the set of bad possible outputs $k$ into sub-intervals of exponentially increasing size as follows. For $i \geq 1$, let

$$
\begin{aligned}
R_i^- &= [k^* - 2^i\alpha, k^* - 2^{i-1}\alpha) \\
R_i^+ &= (k^* + 2^{i-1}\alpha, k^* + 2^i\alpha] \\
R_i &= R_i^- \cup R_i^+
\end{aligned}
$$

Then for any range-specific thresholds $t_i$ for $i \geq 1$, we will still bound the probability of the bad event as follows:

$$
\Pr[|\tilde{k} - k^*| > \alpha] \leq \sum_{i \geq 1} \Pr[\max_{k \in R_i}\{\ell(k) - \ell(k^*)\} > -t_i] + \sum_{i \geq 1} \Pr[\max_{k \in R_i}\{Z_k - Z_{k^*}\} \geq t_i] \quad (21)
$$

We will re-define $U_j$ to denote the i.i.d random variables with mean zero by the alternative log-likelihood, and $S_m$ to denote the sum of $m$ i.i.d $U_j$ as follows:

$$
U_j = \begin{cases} -\log \frac{2P_0(x_j)}{(P_0+P_1)(x_j)} + D_{KL}(P_0||\frac{P_0+P_1}{2}), & j < k^* \\ -\log \frac{2P_1(x_j)}{(P_0+P_1)(x_j)} + D_{KL}(P_1||\frac{P_0+P_1}{2}), & j \geq k^* \end{cases}
$$

$$
S_m = \begin{cases} \sum_{k^*+m \leq j < k^*} U_j, & m < 0 \\ \sum_{k^* \leq j < k^*+m} U_j & m > 0 \end{cases}
$$

With these random variables, we can bound the empirical log-likelihood difference between the true change-point $k^*$ and any candidate $k$ by

$$
\frac{1}{2}[\ell(k) - \ell(k^*)] = \sum_{j=k}^{k^*} \log \frac{P_1(x_j)}{P_0(x_j)} \leq \begin{cases} \sum_{j=k}^{k^*-1} U_j - (k^* - k)D_{KL}(P_0||\frac{P_0+P_1}{2}), & k < k^* \\ \sum_{j=k^*}^{k-1} U_j - (k - k^*)D_{KL}(P_1||\frac{P_0+P_1}{2}), & k \geq k^*. \end{cases}
$$

Then we bound each term in the first set of terms in (21) for any $i \geq 1$ and threshold $t_i = 2^{i-1}\alpha C_M$ as follows:

$$
\Pr[\max_{k \in R_i}\{\ell(k) - \ell(k^*)\} > -2^{i-1}\alpha C_M]
$$

$$
\leq \Pr[\max_{k \in R_i^-}\{\sum_{j=k}^{k^*-1} U_j - (k^* - k)D_{KL}(P_0||\frac{P_0+P_1}{2}))\} > -2^{i-2}\alpha C_M]
$$

$$
+ \Pr[\max_{k \in R_i^+}\{\sum_{j=k^*}^{k-1} U_j - (k - k^*)D_{KL}(P_1||\frac{P_0+P_1}{2})\} > -2^{i-2}\alpha C_M]
$$

$$
\leq \Pr[\max_{k \in [2^{i-1}\alpha]} |S_{-k}| > 2^{i-2}\alpha C_M] + \Pr[\max_{k \in [2^{i-1}\alpha]} |S_k| > 2^{i-2}\alpha C_M] \quad (22)
$$

$$
\leq \frac{4\exp\left(-\frac{2^{i-4}\alpha C_M^2}{C_M+32}\right)}{1 - 2\exp\left(-\frac{2^{i-4}\alpha C_M^2}{C_M+32}\right)} \quad (23)
$$

$$
\leq 8\exp\left(-\frac{2^{i-4}\alpha C_M^2}{C_M + 32}\right) \quad (24)
$$

where (23) follows from an application of Corollary 2 with $\lambda_1 = \lambda_2 = 2^{i-3}\alpha C_M$ and $v = 4$. To apply Corollary 2, we first need to check the conditions of Bernstein inequality. We shall show that for any $j$,

$$
\mathbb{E}\left[\exp(|U_j|) - 1 - |U_j|\right] \leq 2, \quad (25)
$$

and then all conditions of Bernstein inequality are fulfilled. To prove this, let $Y_j$ be the i.i.d. alternative log-likelihood ratio as follows:

$$Y_j = \begin{cases} -\log \frac{2P_0(x_j)}{(P_0+P_1)(x_j)}, & j < k^* \\ -\log \frac{2P_1(x_j)}{(P_0+P_1)(x_j)}, & j \geq k^* \end{cases}$$

Then it suffices to note that

$$\begin{aligned} \mathbb{E}\left[\exp(|U_j|)\right] &= \mathbb{E}\left[\exp(|\log Y_j - \mathbb{E}\left[\log Y_j\right]|)\right] \\ &\leq \mathbb{E}\left[\exp(\log Y_j - \mathbb{E}\left[\log Y_j\right])\right] + \mathbb{E}\left[\exp(\mathbb{E}\left[\log Y_j\right] - \log Y_j)\right] \\ &= \mathbb{E}\left[Y_j\right] e^{-C_M} + \frac{\mathbb{E}\left[1/Y_j\right]}{e^{-C_M}} \\ &\leq e^{-C_M} + \frac{2}{e^{-C_M}}, \end{aligned}$$

and the fact that $e^{-C_M} \in [1, 2]$.

It follows from direct calculations that the condition $\alpha \geq \frac{262}{C_M^2} \log \frac{128}{3\beta} > 363/C_M^2$ implies $2\exp\left(-\frac{2^{i-4}\alpha C_M^2}{C_M+32}\right) < 1/2$, which is used to simplify the denominator as in (24).

We now consider the sum of these terms over all $i$, which will be needed for the final bound on Equation (21).

$$\sum_{i \geq 1} \Pr[\max_{k \in R_i}\{\ell(k) - \ell(k^*)\} > -2^{i-2}\alpha C_M] \leq \frac{16\exp\left(-\frac{2^{-3}\alpha C_M^2}{C_M+32}\right)}{1 - \exp\left(-\frac{2^{-3}\alpha C_M^2}{C_M+32}\right)}$$
$$\leq \frac{64}{3}\exp\left(-\frac{\alpha C_M^2}{262}\right). \tag{26}$$

The first term in (11) in the theorem statement ensures that the expression above is bounded by $\beta/2$, as is required for the private algorithm.

For non-private MLE, we bound each term in the first set of terms in (21) for any $i \geq 1$ and threshold $t_i = 0$ as follows:

$$\Pr[\max_{k \in R_i}\{\ell(k) - \ell(k^*)\} > 0]$$
$$\leq \Pr[\max_{k \in R_i^-}\{\sum_{j=k}^{k^*-1} U_j - (k^* - k)D_{KL}(P_0||\frac{P_0+P_1}{2})\} > 0]$$
$$+ \Pr[\max_{k \in R_i^+}\{\sum_{j=k^*}^{k-1} U_j - (k - k^*)D_{KL}(P_1||\frac{P_0+P_1}{2})\} > 0]$$
$$\leq \Pr[\max_{k \in [2^{i-1}\alpha]}|S_{-k}| > 2^{i-1}\alpha C_M] + \Pr[\max_{k \in [2^{i-1}\alpha]}|S_k| > 2^{i-1}\alpha C_M] \tag{27}$$
$$\leq \frac{4\exp\left(-\frac{2^{i-3}\alpha C_M^2}{C_M+8}\right)}{1 - 2\exp\left(-\frac{2^{i-3}\alpha C_M^2}{C_M+8}\right)} \tag{28}$$
$$\leq 8\exp\left(-\frac{2^{i-3}\alpha C_M^2}{C_M+8}\right) \tag{29}$$

where (28) follows from an application of Corollary 2 with $\lambda_1 = \lambda_2 = 2^{i-2}\alpha C_M$ and $v = 4$. It follows from direct calculations that the condition $\alpha \geq \frac{67}{C_M^2} \log \frac{128}{3\beta} > 92/C_M^2$ implies $2\exp\left(-\frac{2^{i-3}\alpha C_M^2}{C_M+8}\right) < 1/2$, which is used to simplify the denominator as in (29). Then, we consider the sum of these terms over all $i$.

$$\sum_{i\geq 1}\Pr[\max_{k\in R_i}\{\ell(k)-\ell(k^*)\}>0]\leq\frac{16\exp\left(-\frac{2^{-2}\alpha C_M^2}{C_M+8}\right)}{1-\exp\left(-\frac{2^{-2}\alpha C_M^2}{C_M+8}\right)}$$

$$\leq\frac{64}{3}\exp\left(-\frac{\alpha C_M^2}{67}\right). \tag{30}$$

For $\alpha$ as in (10) in the theorem statement, the expression above is bounded by $\beta$, completing the accuracy proof for the non-private MLE.

The calculations for the probability bounds for the Laplace noise terms are the same as those in Theorem 2 with $C$ substituted by $2C_M$, which ends up with a probability no more than another $\beta/2$ under the condition $\alpha\geq\frac{2A\log(16/\beta)}{C_M\epsilon}$.

By Inequality (21), this guarantees that $\Pr[|\tilde{k}-k^*|>\alpha]\leq\beta$ for the assumed ranges of $\alpha$ captured in Equation (11) in the theorem statement, completing the proof. $\qquad\square$

## D  Proofs for ONLINEPCPD

**Theorem 6.** *For hypotheses $P_0, P_1$ such that $\Delta(\ell)<\infty$, a stream of data points $X$ with starting size $n$ drawn from $P_0, P_1$ with true change time $k^*\geq n/2$, privacy parameter $\epsilon>0$, and threshold $T\in[T_L,T_U]$ with*

$$T_L \quad:=\quad 2A\sqrt{2\log\frac{64k^*}{\beta}}-C+\frac{16A}{\epsilon}\log\frac{8k^*}{\beta},$$

$$T_U \quad:=\quad \frac{nC}{2}-\frac{A}{2}\sqrt{n\log(8/\beta)}-\frac{16A}{\epsilon}\log\frac{8k^*}{\beta},$$

*we have that* ONLINEPCPD$(X, P_0, P_1, \epsilon, n, T)$ *is* $(\alpha,\beta)$ *accurate for any $\beta>0$ and*

$$\alpha=\max\left\{\frac{16A^2}{C^2}\log\frac{32n}{\beta},\frac{4A}{C\epsilon}\log\frac{8n}{\beta}\right\}.$$

*In the above expressions, $A=\Delta(\ell)$ and $C=\min\{D_{KL}(P_0||P_1),D_{KL}(P_1||P_0)\}$.*

*Proof.* We first give a range $[T_L, T_U]$ of thresholds that ensure that except with probability $\beta/4$, the randomly sampled data stream satisfies the following two conditions:

1. For $T\geq T_L$, $\max_{k\in[j-n+1,j]}\ell(k)<T-\alpha'$ for every $j<k^*$.

2. For $T\leq T_U$, $\max_{k\in[k^*-n/2,k^*+n/2)}\ell(k)>T+\alpha'$.

When these conditions are satisfied, the ABOVETHRESH guarantee ensures that except with probability $\beta/4$, the randomness of the online algorithm ensures that it calls the offline algorithm on a window of data containing the true change-point. Then we will argue that our overall accuracy follows from the offline guarantee, where we will allow failure probability $\beta/2$.

We will get the first condition by taking a union bound over all windows tested before the change point of the probability that the maximum log-likelihood $\max_k\ell(k)$ for $n$ elements $X=(x_1,\dots,x_n)$ sampled from $P_0$ exceed a given threshold. To bound this probability, we first define the following random variables.

$$U_j=-\log\frac{P_0(x_j)}{P_1(x_j)}+D_{KL}(P_0||P_1)\qquad\qquad S_m=\sum_{1\leq j\leq m}U_j$$

We note that each $\ell(k)$ is the sum of i.i.d. random variables, and that the maximum log-likelihood over $m$ consecutive elements is equal in distribution to $\max_{k\in[m]}S_k-kD_{KL}(P_0||P_1)$. This yields the

first inequality below. Inequality (31) comes from applying Corollary 1 with $\lambda_1 = \lambda_2 = 2^{i-2}C + t/2$ and interval length $L = A$.

$$\Pr\left[\max_{1 \leq k \leq n} \ell(k) > t\right] \leq \sum_{i \geq 1} \Pr[\max_{k \in [2^{i-1}, 2^i)} \{S_k - kD_{KL}(P_0||P_1)\} > t]$$

$$\leq \sum_{i \geq 1} \Pr[\max_{k \in [2^{i-1}]} S_k > 2^{i-1}C + t]$$

$$\leq \sum_{i \geq 1} \frac{2\exp(-(2^{i-2}C + t/2)^2/(2^{i-2}A^2))}{1 - 2\exp(-(2^{i-2}C + t/2)^2/(2^{i-2}A^2))} \tag{31}$$

$$\leq 4 \sum_{i \geq 1} \exp(-(2^{i-2}C + t/2)^2/(2^{i-2}A^2)) \tag{32}$$

$$\leq 8 \exp(-(2^{-1}C + t/2)^2/(2^{-1}A^2)) \tag{33}$$

Inequalities (32) and (33) follow by plugging in $t = 2A\sqrt{2\log\frac{64k^*}{\beta}} - C$. This ensures that $1 - 2\exp(-(2^{i-2}C + t/2)^2/(2^{i-2}A^2)) \geq 1/2$, giving Inequality (32), and that the series is increasing exponentially in $i$, so we can collapse the sum with another factor of 2 by considering only $i = 1$ as in Inequality (33). This value of $t$ also ensures that the bound of Inequality (33) is at most $\beta/(8k^*)$. Taking the union bound over all the windows prior to the change-point, this shows that Condition 1 holds for $T_L = 2A\sqrt{2\log\frac{64k^*}{\beta}} - C + \alpha'$ except with probability $\beta/8$.

To show that the second condition holds except with additional probability $\beta/8$, we consider the window of data with the first half of data drawn from $P_0$ and the second half drawn from $P_1$ and bound the probability that $\ell(k^*)$ in this window is less than a given threshold as follows. We note that $\ell(k^*)$ is the sum of $n/2$ i.i.d. random variables, so we define mean-zero random variables $V_j = -\log\frac{P_1(x_j)}{P_0(x_j)} + D_{KL}(P_1||P_0)$ and bound their sum using Hoeffding's inequality:

$$\Pr[\max_{k^* - n/2 \leq k < k^* + n/2} \ell(k) < t] \leq \Pr[\ell(k^*) < t]$$

$$\leq \Pr[\sum_{k \in [n/2]} V_j > nC/2 - t]$$

$$\leq \exp(-4(nC/2 - t)^2/(nA^2)) \tag{34}$$

Plugging in $t = \frac{nC}{2} - \frac{A}{2}\sqrt{n\log(8/\beta)}$ in this final expression ensures that $(34) \leq \beta/8$. This ensures that Condition 2 is satisfied except with probability $\beta/8$ for $T_U = nC/2 - A\sqrt{2\log(8/\beta)} - \alpha'$.

Then we can instantiate the ABOVETHRESH accuracy guarantee with privacy parameter $\epsilon/2$ and accuracy parameter $\beta/4$ to ensure that for $\alpha' = \frac{16A\log(8k^*/\beta)}{\epsilon}$ when Conditions 1 and 2 are satisfied, ABOVETHRESH will identify a window containing the true change-point except with probability $\beta/4$. Combining this with the $\beta/4$ probability that Conditions 1 and 2 fail to hold when $T \in [T_L, T_U]$, we get that ONLINEPCPD calls OFFLINEPCPD in a window containing the change-point except with probability $\beta/2$ over the randomness of the data and of the online portion of the algorithm.

We next instantiate OFFLINEPCPD with appropriate parameters to ensure that conditioned on being called in the correct window, it will output a $\tilde{k}$ that is within $\alpha$ of the true change-point $k^*$ with probability at most $\beta/2$. We can then complete the proof by taking a union bound over all the failure probabilities.

Our offline accuracy guarantee requires data points sampled i.i.d. from $P_0$ before the change point and from $P_1$ thereafter, so it remains to show that conditioning on the event that we call the offline algorithm in a correct window does not harm the accuracy guarantee too much. For a window size $n$, change-point $k^*$, stream $X$ of at least $k^* + n/2$ data points, set of random coins required by ONLINEPCPD and its call to OFFLINEPCPD, and a stopping index $k > n/2$, let $N(k)$ denote the event that ONLINEPCPD calls OFFLINEPCPD on a window centered at $k$, and let $F(k)$ denote the event that OFFLINEPCPD on the window centered at $k$ fails to output an approximation within $\alpha$ of $k^*$.

Our previous argument bounds the probability of all $N(k)$ for $k$ outside of a good range $G = (k^* - n/2, k^*]$, and our offline guarantee bounds the probability of $F(k)$ for any $k \in G$ as long as the data are drawn according to the change-point model. Then the overall probability of a bad event can be bounded as follows, where the probability is over the $X$ drawn from $P_0$ and $P_1$ with change-point $k^*$ and of the randomness of the algorithm:

$$\Pr[|\tilde{k} - k^*| > \alpha] = \sum_{k > n/2} \Pr[N(k) \cap F(k)]$$

$$\leq \sum_{k \notin G} \Pr[N(k)] + \sum_{k \in G} \Pr[F(k)]$$

The first summation is at most $\beta/2$ by our previous arguments. By instantiation of Theorem 2 for OFFLINEPCPD with a $\beta/(2n)$ and $\epsilon/2$, the second summation is also bounded by $\beta/2$ when $\alpha = \max\{\frac{32A^2}{C^2} \log \frac{64n}{\beta}, \frac{8A}{C\epsilon} \log \frac{16n}{\beta}\}$. □

## E  Empirical validation of ONLINEPCPD

We also run Monte Carlo simulations of our online change-point detection algorithm ONLINEPCPD, when the data points arrive sequentially and the true change occurs at time $k^* = 5000$. We choose the appropriate threshold $T$ by setting a constraint that an algorithm must have positive and negative false alarm rates both at most $0.1$. The range of threshold $T$ for the online algorithm needs to be non-empty, which impacts our choice of sliding window size $n$. Unfortunately the window size of $n = 200$ used in the offline simulations is not sufficient for our online examples. A larger window size is needed to detect smaller changes or under higher levels of noise. For this reason, we choose window size $n = 700$ and restrict our online simulations to the large change scenario (A) and privacy parameters $\epsilon = 0.5, 1, \infty$.

For the online simulations, we use several key ideas in Section 4 to speed up the numerical search of the threshold $T$. On the one hand, the threshold $T$ cannot be too small, otherwise a false alarm will be likely. To control the false alarm rate of $0.10$ with up to $k^* = 5000$ sliding windows, a conservative lower bound of the threshold $T$ is the $1 - 0.10/5000 = 0.99998$ quantile of the noisy versions of $W_n$ with $n = 700$ under the *pre-change* distribution. On the other hand, the threshold $T$ cannot be too large, otherwise it will fail to detect a true change in any sliding windows of size $n = 700$. A useful upper bound of the threshold $T$ is the $10\%$ quantile of the noisy versions of CUSUM statistics $W_n = \max_{1 \leq k \leq n} \ell_k$ with $n = 700$ when the change occurs at time $350$, since it will guarantee that the online algorithms raise an alarm with probability at least $0.9$ during the time interval $[4650, 5350]$.

Next, we simulate $10^6$ realizations of the CUSUM statistics $W_n = \max_{1 \leq k \leq n} \ell_k$ with $n = 700$ in both the pre-change and post-change cases. In each case, we speed up the computation of $W_i$ by using the recursive form $W_i = \max W_{i-1}, 0 + \log(P_1(X_i)/P_0(X_i))$ for $i \geq 1$. The empirical quantiles of the noisy versions of $W_n$ with $n = 700$ under the pre- and post- change cases will yield the lower and upper bounds of the threshold $T$. When the range of the threshold $T$ is non-empty, we choose one that is closest to the upper bound. For the Bernoulli model, we use $T = 220$ for all values of $\epsilon = 0.5, 1, \infty$. In the Gaussian model, our window size $n = 700$ is not sufficient to ensure non-empty range of $T$ under false alarm rate $0.2$ for $\epsilon = 0.5, 1$, so we relax the false alarm constraints for these $\epsilon$ values and choose $T = 180, 150, 100$ for $\epsilon = 0.5, 1, \infty$, respectively. Figure 2 (c) indeed shows that the false alarm rates are high in the Gaussian model with $\epsilon = 0.5, 1$.

Figure 2 summarizes our online simulations results for both Bernoulli and Gaussian models using a sliding window size $n = 700$ to detect a large change (scenario A) that occurs at time $k^* = 5000$. Suppose our online algorithm raises an alarm at time $j$ with the estimated change-point $\tilde{k}_j$ for the sliding window of the observations, $\{x_{j-n+1}, \cdots, x_j\}$. Two probabilities are plotted: one is the marginal probability of inaccurate estimation and false alarm, $\beta_1 = \Pr(|\tilde{k}_j - k^*| > \alpha$ or $k^* \notin (j - n + 1, j))$, and the other is the conditional probability of inaccurate estimation conditioned on raising an alarm correctly, $\beta_2 = \Pr(|\tilde{k}_j - k^*| > \alpha | j - n + 1 \leq k^* \leq j)$. As $\alpha \to \infty$, the probability $\beta_1$ becomes the false alarm rate plus the error rate related to the Laplace noise in hypothesis testing. For both Bernoulli and Gaussian models, the right-hand side plots in Figure 2 (b and d) suggest that the online accuracy conditioned on correctly raising an alarm is very similar to the offline accuracy. Our plots show that the primary challenge in the online setting is determining when to raise an alarm

in a sequence of sliding windows of observations. Once such window is identified correctly, the offline estimation algorithm can be used to accurately estimate the change-point.

(a) Bernoulli, inaccurate estimation and false alarm

(b) Bernoulli, inaccurate estimation conditioned on no false alarm

(c) Gaussian, inaccurate estimation and false alarm

(d) Gaussian, inaccurate estimation conditioned on no false alarm

Figure 2: Probability of inaccurate estimation and false alarm (left) and probability of accurate report conditioned on raising an alarm correctly (right) for Monte Carlo simulations with Bernoulli and Gaussian data. Each simulation involves $10^6$ runs of ONLINEPCPD with window size $n = 700$ and varying $\epsilon$ on data generated by i.i.d. samples from appropriate distributions with change point $k^* = 5000$.