[Reviews · NeurIPS 2018]

Reviewer 1



This paper studies differentially-private change-point detection -- detecting the point at which samples from a distribution stop coming from P_0, and instead start coming from P_1. It provides the first finite sample guarantees for the MLE for this problem non-privately (I'm not an expert in change-point detection, can anyone else confirm? It seems natural enough that I would expect someone else to have studied it prior...). The authors study the offline problem (where the entire stream is given at once), as well as the online problem. I like this paper largely because it initiates the study of what I consider to be a fairly basic and interesting question, private change-point detection. My main criticism of the work is the assumptions that are required on the data. In particular, there is only a true differential privacy guarantee in the case where the likelihood ratio is bounded. In the case where it is bounded only with high probability, there is a data-dependent privacy guarantee, which makes me uneasy (what if the model is wrong, who knows where the data truly comes from, etc.). There is some vague text justification on why a bound is necessary -- is it possible to make this justification rigorous, in the form of a theorem? The algorithmic techniques are not too surprising, but that's acceptable in my opinion for the first paper is the area. The authors should note that the submission format is for 8 pages + citations, with no appendices (which should be put in the supplementary material). EDIT: The authors are advised to compare their results with the recent line of works on private hypothesis testing, particularly the ones focusing on minimax sample complexities, as these seem to focus on very similar problems (see, i.e. Gaboardi et al.'16, Kifer-Rogers'17, Cai et al.'17, Aliakbarpour et al.'18, Acharya et al.'17, etc.).

Reviewer 2



This paper studies how to detect a change in a sequence of data while satisfying differential privacy. In this problem, a sequence of n data points is split into points x_1, x_2, . . ., x_{k-1} drawn iid from known distribution P_0 and points x_k, x_{k+1}, . . ., x_n drawn iid from known distribution P_1, and the goal is to recover (a value close to) the unknown k. The paper first considers the offline case where all n points are known to the algorithm in advance. In the first subcase, they assume P_0 and P_1 have a uniform bound on log(P_1(x)/P_0(x)) from both above and below. Under this assumption, they prove a finite-sample accuracy guarantee for a non-private MLE and a similar guarantee for the private version. In the second subcase, they relax this uniform assumption to one that bounds the log probability ratio everywhere except for a region of small probability mass and obtain slightly worse guarantees. Both algorithms adapt a well-known change-point detection technique and get privacy by adding a very simple (and natural) differentially private protocol on top of it, Report Noisy Max. The more difficult online case is correspondingly more complex. The algorithmic idea is simple enough -- use an existing sparse vector-type algorithm to track whether a change is likely, then call the offline method to compute an estimate on the data so far. This setting appears to require the same uniform assumption as subcase 1 for the offline version and obtains a similar-looking guarantee except for a log dependence on the number of points seen, and some requirements on the threshold T. Finally, the paper conducts some simple experiments for the case when P_0 and P_1 are both Bernoulli. These largely bear out the theory. I did not check the proofs in the Appendix. I'm familiar with the differential privacy literature and not familiar with the change point detection literature. Positive points (no order): This paper provides, to the best of my knowledge, the first differentially private solution for change-point detection. Its algorithms are simple and intuitive from a differential privacy perspective, and the accuracy guarantees "look right". The fact that these guarantees are (taking the paper's claim here at face value) the first finite-sample accuracy guarantees for change-point detection, even absent privacy, is a nice contribution as well. The paper is generally well-written and easy to follow. Negative points (in order of decreasing importance): 1. As stated above, I'm not very familiar with the change point detection literature, but looking at a few papers (e.g. https://arxiv.org/pdf/1306.4933.pdf) suggests that "change point detection" often refers to something much more general than the problem presented here, where it is assumed that only one change takes place and both the before distribution P_0 and after distribution P_1 are known. Similarly, the results presented require further assumptions (e.g. the bounds on A) to hold. This is a bit disappointing given the generality of the original claims ("provide the first accuracy guarantees for the MLE from a finite sample"). Signposting these restrictions earlier would help. Similarly, the language in lines 109-111 suggests that the algorithms should be able to identify problem instances where *no* change occurs, but this doesn't seem to be the case. Along the same lines, I think the paper is a little too blithe about the assumptions on delta(ell) and A_delta. The assumption on delta(ell) almost looks to me like a stealth smoothness assumption (neither distribution is "spiky"). Examples of P_0 and P_1 and their delta(ell) and A_delta (so, an expansion of lines 167-8 to other distributions) would be nice. 2. The online section (4) feels under-explained. In particular, there is little guidance on actually choosing T -- the T_L and T_U in Theorem 6, being defined in terms of k*, don't help. There appears to be some information on this in Appendix D -- a paragraph or two to this effect in the main body would he very helpful. Even if choosing T is actually hard in some way, it's good to be honest about it. Right now it makes me suspicious. 3. The experiments don't seem very useful to me. The Bernoulli case is close to the easiest P_0 and P_1 possible, and the most interesting case (misspecified change) still tests a fairly small error (misspecifying P_1 as Bernoulli 0.4 instead of 0.8, which is even easier to detect). I have similar misgiving about the Gaussian case -- both seem like "toy" distributions" that I would expect good performance on. It would be useful to see distributions for which the algorithms perform poorly, or at least less well-behaved or noisy distributions. I think as is the page this section takes up would be better spent on some of the other negative points mentioned here. This complaint goes hand in hand with my misgivings about bounding delta(ell) and A_sigma -- are these assumptions so strong that only very "nice" distributions satisfy them? 4. Minor points: should be "extend our offline results to the online setting" in lines 79-80, "the a single entry" in line 217. Overall: This paper offers the first differentially private solution to (a subclass of) change point detection problems. Subject to certain assumptions, it obtains accuracy bounds that depend polynomially on "sensitivity" properties of the generating distributions involved, for both offline and online cases. I think these dependencies are for the most part necessary for a (nontrivial) differentially private solution, but I think the paper suffers from insufficient explanation of what A, C, A_delta, C_M, etc. actually look like (points 1 and 3). Similarly, the online algorithm has a fuzzy dependence (in terms of actually running the thing, not the accuracy guarantee) on T (point 2). For these reasons, I vote marginal accept. However, I think many of the problems I've noted are not too hard to solve, unless the assumptions are actually very strong and clarifying them reveals the weakness of the paper (which I don't expect to be true). I therefore look forward to the authors' rebuttal. Sample text addressing the concerns above would be a plus. ---Update After Response--- After reading the authors' response, I have kept my score at a 6. This is not because the response was bad -- it clarified several of my questions. However, I was less convinced by responses to the questions of other authors, particularly Reviewer 1's concerns about how data-dependent the privacy guarantee is, which I seem to have missed on my first read. I think as is the paper is not very clear about this and the response does not help. This is a drawback for a differential privacy paper, even if it's otherwise a promising extension of differential privacy to an interesting problem. I just feel that the current version of the paper is not quite ready enough to merit a clear accept for me.

Reviewer 3



This paper proposes differentially private change-point detection methods for both offline and online settings.The offline (easier) case is handled with report noisy max mechanism, while the online case uses above threshold mechanism to find a likely window for a change-point, then uses the offline algorithm on that window. The report noisy max mechanism for the offline case requires finite likelihood ratios (bounded sensitivity). When this is the case, for example the uniform distribution case, the resulting method is epsilon-DP. The authors provide (alpha,beta)-accuracy analysis for this case. They also relax this by allowing infinite likelihood ratio with delta probability, which leads to an (epsilon,delta)-DP method. The accuracy bounds are weakened in this case also. A well-written paper with a clear message. The impact of its contribution is limited.